# Turath-150K: Image Database of Arab Heritage

**Dani Kiyasseh**
Department of Engineering Science
University of Oxford
Oxford, UK
dani.kiyasseh@eng.ox.ac.uk

**Rasheed El-Bouri**
Department of Engineering Science
University of Oxford
Oxford, UK
rasheed.el-bouri@eng.ox.ac.uk

## Abstract

Large-scale image databases remain largely biased towards objects and activities encountered in a select few cultures. This absence of culturally-diverse images, which we refer to as the "hidden tail", limits the applicability of pre-trained neural networks and inadvertently excludes researchers from under-represented regions. To begin remedying this issue, we curate Turath-150K, a database of images of the Arab world that reflect objects, activities, and scenarios commonly found there. In the process, we introduce three benchmark databases, Turath Standard, Art, and UNESCO, specialised subsets of the Turath dataset. After demonstrating the limitations of existing networks pre-trained on ImageNet when deployed on such benchmarks, we train and evaluate several networks on the task of image classification. As a consequence of Turath, we hope to engage machine learning researchers in under-represented regions, and to inspire the release of additional culture-focused databases. The database can be accessed here: `danikiyasseh.github.io/Turath`.

## 1   Introduction

Deep neural networks have exhibited great success in performing various computer vision tasks, such as image classification [1], object detection [2], and segmentation [3]. One of the key factors and driving forces behind the success of such networks is access to large-scale, annotated datasets that consist of samples that are mostly representative of the underlying data distribution. To that end, publicly-available datasets, such as ImageNet [4], SUN [5], and Places [6], attempt to capture a diverse set of images that are reflective of objects and scenarios encountered "in the wild". Such images typically belong to categories guided by the WordNet hierarchy [7] and which are diversified by incorporating various adjectives into search queries (e.g., night, foggy, etc.)

Despite these efforts, existing databases remain largely biased towards objects, activities, and scenarios commonly encountered in a small subset of cultures [8], define "diversity" narrowly, and do not account for the long-tail of image categories that are common in other cultures. For example, items and activities common in other parts of the world, such as those in the Arab world, are under-represented, if at all, in existing image databases [9]. Examples include traditional daily clothing items, such as the "thobe", and sporting activities, such as falconry. We refer to these under-represented categories, in which *no* images are available in existing databases, as the "hidden tail". This is analogous to the "long tail" of image categories, in which *few* images are available, that the machine learning community has dedicated substantial effort to better representing.

Such an exclusion of culturally-diverse images has a technical, societal, and ethical impact on the machine learning community. From a technical perspective, the absence of diverse images in existing databases violates the assumption that samples are from "the wild" and representative of the underlying data distribution. By evaluating networks on such narrow samples, their performance tends to be an over-estimate. Moreover, culturally-diverse image categories are effectively out-of-

35th Conference on Neural Information Processing Systems (NeurIPS 2021), Sydney, Australia.

distribution (OOD) samples notorious for degrading the performance of trained networks [10], a phenomenon shown to be more prominent when transferring across geographical regions [11]. On a societal level, pre-trained networks are less likely to be of direct value to researchers residing in, or operating with, under-represented communities. This is driven by the poor performance of such networks on OOD samples, which is a direct consequence of the cultural bias inherent in the datasets used to train such networks. With this imbalance in the applicability of networks across cultures, under-represented communities are unlikely to capture the benefits of computer vision-based advancements. Furthermore, the machine learning community's lack of exposure to data from diverse cultures suggests that researchers have less of an opportunity to learn about such cultures. Such dataset-based learning, the acquisition of skills and knowledge via datasets, has been evident with, for example, the Caltech-UCSD Birds 200 database [12] and ornithology. On an ethical level, the absence of data to which researchers can relate implicitly excludes these researchers from more actively engaging with the machine learning community. As such, it is to the advantage of the community to build the infrastructure that incentivizes the involvement of practitioners from a more diverse background in machine learning.

In this work, we aim to increase the cultural diversity of images that are available for training neural networks. Hence, we present the Turath-150K[1] database, a large-scale dataset of images depicting objects, activities, and scenarios that are rooted in the Arab world and culture. We chose this culture as an exemple, particularly due to its under-representation in existing publicly-available datasets, and hope other researchers follow suit with publishing datasets depicting cultures from around the globe. Specifically, our contributions are the following: (1) we build a large-scale database of images, entitled Turath-150K, the first of its kind that centres around life in the Arab world. For benchmarking purposes, we split the database into three distinct subsets; Turath-Standard, Turath-Art (focusing on art from the Arab world), and Turath-UNESCO (focusing on heritage sites located in the Arab world). (2) We shed light on the limitations of deep neural networks pre-trained on ImageNet by showing that they are unable to deal with the out-of-distribution samples of the Turath database. (3) We evaluate various networks on the Turath benchmark databases and demonstrate their image classification performance on both high and low-level categories.

## 2 Related work

There exists a multitude of publicly-available image databases that have been exploited for the training of deep neural networks. We outline several that we believe are most similar to our work and also elucidate how our database, Turath, differs significantly in motivation, scope, and content.

**Scene recognition databases** The task of scene recognition involves identifying scenes based on images. To facilitate achieving this task, the SUN397 database [5] was designed to contain 100K images of 397 scenes. The vast majority of these scene categories are motivated by the WordNet hierarchy [7]. Similarly, the Places database [6] was designed to contain 2.5 million images of 365 high-level scenes, such as coffee-shop, nursery, and train station. Although extensive in terms of the number of samples, the scene categories lack the granularity that we offer and do not trivially extend to the Arab world. Moreover, Turath is not exclusively limited to scenes (see Sec. 3) and goes beyond the narrow WordNet hierarchy by explicitly accounting for entities in the Arab world.

**Object classification databases** The task of object classification focuses on identifying object(s) in an image. To propel research on this front, the Caltech 256 database [13] was designed to contain 30K images of everyday objects, such as cameras and laptops. The COCO database [14] is much more extensive with 330K images corresponding to 80 object categories and consisting of multiple annotations, including segmentation maps at various levels of detail. Nonetheless, such databases differ in motivation, scope, and content from our database. In order to increase the cultural diversity of datasets, we turn our attention to objects, activities, and scenarios commonly found in the Arab world. Moreover, our image annotations are not only absent from existing databases but also offer a finer resolution of class label. We explain this in further depth in the next section.

**Out-of-distribution databases** Researchers have adopted various approaches to handle the generalization of their models to out-of-distribution samples. These approaches can be split according

---

[1]Turath roughly means heritage in Arabic

to whether they are implemented during training or evaluation, with the latter being more relevant to our work. For example, ImageNet-R [11] is an evaluation database of 30K images, spanning 200 ImageNet categories, rendered in different styles and textures. While their approach augments existing ImageNet categories, our database includes image samples from categories *beyond* the ImageNet-1K. ImageNet-O [10] is an evaluation database that claims to reflect label distribution shift, yet still only comprises images from 200 categories in ImageNet-1K. Whereas ImageNet-O is focused on evaluating out-of-distribution detectors, the Turath database is primarily focused on increasing the representation of image categories that are under-represented in ImageNet.

# 3  Design and construction of the Turath database

In light of our emphasis on increasing the cultural diversity of images, we aimed to construct a database that satisfies the following desiderata:

1. **Heritage -** Categories of images must be specific to the cultures of the Arab world; we reiterate that although our particular choice of culture stems from its under-representation in existing publicly-available databases, it is simply an example. There remains a multitude of rich cultures that are under-represented and we hope other researchers eventually publish such culture-specific databases, be they in the form of images, audio, or video.

2. **Quantity -** Each category must contain a sufficient number of images to facilitate learning; although the term "sufficient" is nebulous and category-dependent, existing databases have demonstrated success with at least 50 images per category. We quadruple that amount and aim for at least 200 images per category.

3. **Real World -** Images in each category must reflect those commonly encountered "in the wild"; networks trained on image databases have a number of applications but they are, arguably, most useful when applied in the real world to challenges afflicting stakeholders from patients to farmers. To that end, we aim to collect natural RGB images.

The construction of the Turath database consisted of three main stages. We first defined keywords to guide the download of images from web-based search engines. We then used these keywords to assign images an annotation. Lastly, and as a form of noise reduction, we trained several classifiers to distinguish between categories and removed images that were likely to be associated with the incorrect annotation. We now describe these stages in more depth.

**Stage 1: Defining keywords and downloading the images**  Existing image databases such as ImageNet and Places were created by performing query-based searches using online search engines. In this setting, the choice of queries determines the type and quality of images that are retrieved. In our context, and in contrast to the aforementioned work, the WordNet hierarchy [7] did not satisfy our outlined desiderata. This is primarily because WordNet was not designed for the Arab world and thus does not contain categories that are directly relevant for our purposes. Although an Arabic WordNet [15] does exist, it is unable to capture the cultural focus and the *micro* categories (described next) that we are searching for.

Given our emphasis on the Arab world as an example, we conducted query-based searches of entities engrossed in the diverse cultures of the region. This ranged from categories of images with a low level of detail, such as cities and architecture, to those with a high level of detail, such as traditional food and clothing. Each of these *macro* categories are formed by grouping several *micro* categories. For example, the *macro* category of Cities comprises 25+ *micro* categories of images from specific cities in the Arab world, e.g., Damascus, Cairo, and Casablanca. To emphasize the under-representation of images of these cities in existing databases, we note that the largest image database of cities, World Cities [16], with 2.25M images, covers a single city (Dubai) in the Arab world. In Fig. 1, we present image samples from three macro categories, Dates, Architecture, and Souq, each containing four *micro* categories.

In addition to retrieving images from the categories mentioned above, we dedicate time and effort to curating two additional *macro* categories that comprise a large number of *micro* categories. Specifically, these revolve around Arab Art and United Nations Educational, Scientific and Cultural Organization (UNESCO) sites. When retrieving images that belong to the Arab Art category, we followed the same strategy of query-based searches. However, given the breadth of this field and to

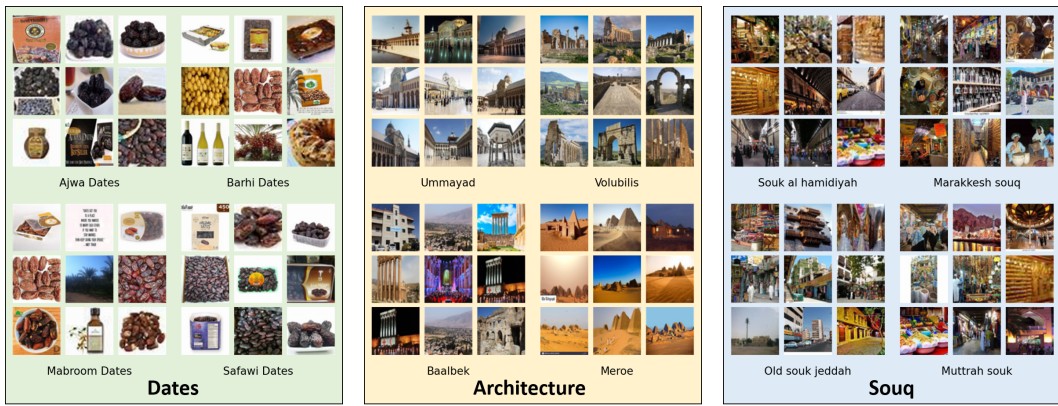

Figure 1: **Images samples from a subset of categories available in Turath.** Four *micro* categories are shown for each of the three *macro* categories, DATES, ARCHITECTURE, and SOUQ. The image categories range from objects with low-level details, such as dates, to locations with high-level details, such as architecture.

keep the task of downloading images tractable and organized, our search queries were based on artists' names. To that end, we identified 425 names available on the Barjeel Art Foundation website[2]. As for the UNESCO category, our search queries were based on the names of 88 recognized UNESCO sites in the Arab world[3].

**Stage 2: Labelling the images using keywords**   Each image in the Turath database has two image-level annotations; a *micro* label and a *macro* label. To assign downloaded images to *micro* categories, we follow the strategy proposed by Marin *et al.* [17] where each category is defined by the query used to search for those images. Similar to their conclusions, we also find that such an approach leads to relatively high quality images that are relevant to the search query. We then grouped *micro* categories with similar themes into *macro* categories. As an example, we grouped seven types of dates (*micro*) into a single DATES category (*macro*).

**Stage 3: Filtering the images with classifier-based labelling**   Despite our effort to conduct searches using queries that are unambiguous and descriptive, upon further inspection, we found that certain categories contained images that were irrelevant. This was most prominent amongst images that belonged to artists. For example, the query inji efflatoun art returned art pieces associated with the artist Inji Efflatoun, as desired, but also images of the artist herself.

To remedy this situation, we exploited the prior knowledge that out-of-distribution (OOD) image samples are likely to be of artists' faces. Therefore, given our emphasis on retaining images of art pieces, we designed a binary classifier that distinguished between images of art and those of faces. To train such a classifier, we needed images with relatively high quality labels. For those in the "art" domain, we grouped all the categories in ImageNet-R [11], which comprises images from ImageNet rendered artistically, into a single category. For those in the "faces" domain, we exploited images from the LFW database [18], which comprises 13K images of faces, and grouped them into a single category. After training this classifier, we performed inference on *our* set of artistic images. Given that the majority of images are those of art pieces, we would expect the distribution of output probabilities to be bi-modal and skewed towards the value zero (i.e., corresponding to art images). This is indeed what we find empirically, as shown in Fig. 2. Upon manual inspection of the images, we chose a threshold value of $0.1$, whereby approximately 26.1% of image samples believed to have been of art are instead identified as a face. These 27,302 images are removed from the database.

Detecting OOD images of human faces exploited the implicit bias that human faces comprised the majority of the OOD images. However, not all OOD images contain human faces. To investigate this, we explored more general approaches involving one-class SVMs [19], deep autoencoding GMMs [20], adversarial networks [21], geometric transformations [22] and self-supervised classification

---

[2]https://www.barjeelartfoundation.org/

[3]https://whc.unesco.org/en/list/&&&order=region

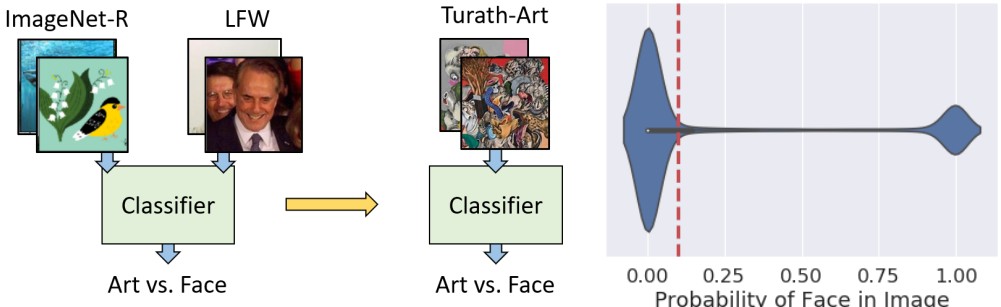

Figure 2: **Pipeline for cleaning data in Turath database. (Left)** Classifier-based cleaning of data. We trained a binary classifier to distinguish between images of art (ImageNet-R) and faces (LFW) and deployed it on Turath-Art. **(Right)** Distribution of probabilities output by binary classifier deployed on all images of Turath Art. We found that, when a threshold of $0.1$ is chosen, approximately 26.1% of images are identified as a face.

networks [23]. We empirically found that although this self-supervised approach was preferable to the remaining methods, it was still unable to reliably identify OOD samples.

## 4 Turath benchmark databases

The Turath database comprises three specialized subsets of data that contain images from mutually-exclusive categories. Hereafter, these subsets will be referred to as Turath Standard, Turath Art, and Turath UNESCO, respectively, and, in this section, will be described in depth. We chose to separate the database along these dimensions to account for the different resolution of the categories, as will be shown next.

**Turath Standard** The Turath Standard benchmark database comprises images reflecting the diverse range of objects, activities, and scenarios commonly encountered in the Arab world. Each image has a *macro* and *micro* image-level category annotation. The twelve macro categories are Cities, Food, Nature, Architecture, Dessert, Clothing, Instruments, Activities, Drinks, Souq, Dates, and Religious Sites. The complete list of the more granular *micro* categories can be found in Appendix A. The number of images in each of these micro categories is presented in Fig. 3a. We can see that each micro category has anywhere between $50 - 500$ images. This is by design since we explicitly searched for *up to* 500 images per category and excluded categories with fewer than 50 images. We applied this strategy to all benchmark databases to avoid categories with too few images which may contain noise and thus hinder a network's ability to learn.

Table 1: **Overview of training, validation, and test splits for the Turath benchmark databases.** The number of macro categories is shown in brackets.

|  | Turath Database | | |
|---|---|---|---|
|  | Standard | Art | UNESCO |
| Training | 38,894 | 46,665 | 9,540 |
| Valid. | 6,418 | 7,531 | 1,558 |
| Test | 19,472 | 22,969 | 4,778 |
| Categories | 269 (12) | 419 | 79 |

For benchmarking, the Turath Standard database contains 38,894 images in the training set, 6,418 images in the validation set, and 19,472 images in the test set (see Table 1). Unless otherwise specified, all data splits are performed uniformly at random with a ratio of 70:10:20 for the training, validation, and test sets, respectively.

**Turath Art** The Turath Art benchmark comprises images of art (e.g., paintings, sculptures, etc.) created by Arab artists alongside annotations, at the image-level, of such artists. We purposefully excluded these categories from the Turath Standard benchmark for the following reasons. First, the large number of *micro* categories (419) that would have fallen under the *macro* category of Art would have overwhelmed the categories outlined in the Turath Standard benchmark. Second, distinguishing between images containing intricate, low-level details reflected by paintings, sculptures, etc., poses a difficult task, in and of itself. As a result, this warranted a

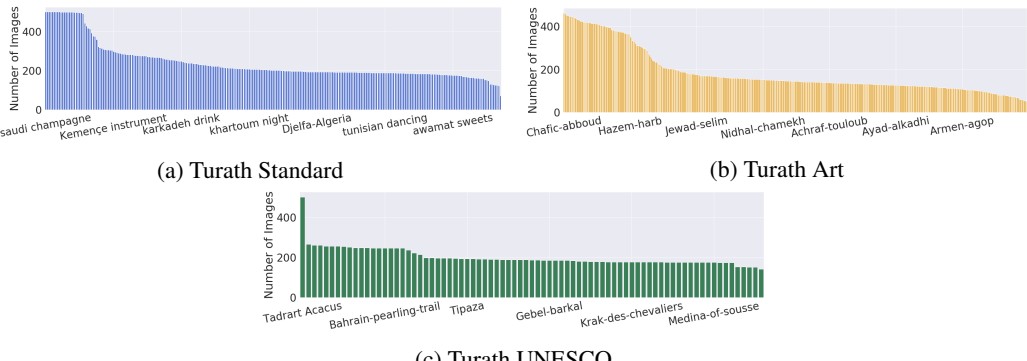

(a) Turath Standard

(b) Turath Art

(c) Turath UNESCO

Figure 3: **Number of images per *micro* category in each of the benchmark databases.** Each micro category contains anywhere between 50-500 images. For clarity, we present only a subset of the micro category names. The full list of categories can be found in Appendix A.

distinct specialized benchmark, which we refer to as Turath Art. In Fig. 3b, we present the number of images in each of the 419 artist categories, and include a subset of the artists' names for clarity. For benchmarking, the Turath Art database contains 38,445 images in the training set, 6,354 images in the validation set, and 19,324 images in the test set.

**Turath UNESCO**   The Turath UNESCO benchmark comprises images of UNESCO world heritage sites in the Arab world alongside annotations, at the image-level, of these sites. We present, in Fig. 3c, the total number of images in each of the 79 categories. For benchmarking, the Turath UNESCO database contains 9,540 images in the training set, 1,558 images in the validation set, and 4,778 images in the test set.

## 5   Experimental results

### 5.1   Limitations of networks pre-trained on ImageNet

The utility of a pre-trained neural network is contingent upon the similarity of the upstream task, on which the network was trained, and the downstream task, on which the network is deployed [24]. To qualitatively evaluate this utility in the context of the Turath database, we randomly sample images from each of the benchmark databases, perform a forward pass through an EfficientNet [25] pre-trained on ImageNet, and compare the Top-5 predictions to the ground-truth label (see Fig. 4). We find that, across the benchmarks, EfficientNet assigns a high probability mass to incorrect image categories. For example, it classified a sculpture by the artist $\mathrm{Maysaloun\ Faraj}$ as an envelope with a confidence score $(0.564)$ and $\mathrm{Gebel\ Barkal}$, pyramids in Sudan, as a seashore with a confidence score $(0.266)$. These results also suggest that confidence-based decisions, such as network classification abstention and out-of-distribution detection [26], may be of little value in this context. We show that these limitations also extend to other neural architectures (see Appendix C).

### 5.2   Image classification on Turath benchmark databases

In this section, we adapt networks pre-trained on ImageNet using data from the Turath database benchmarks. We do so by introducing, and randomly initializing, a classification head, $p_\theta : h \rightarrow \hat{y} \in \mathbb{R}^C$, that maps the penultimate representation, $h$, of the feature extractor network to the predicted probability distribution, $\hat{y}$, over the set of image categories, $C \in \{12, 269, 419, 79\}$ depending on the benchmark database. In the linear evaluation phase, we freeze the parameters of the feature extractor network whereas in the fine-tuning phase, we use those parameters as an initialization and update them accordingly. In both phases, we train networks using the Adam optimizer with a categorical cross-entropy loss and a learning rate, $lr \in [1e^{-3}, 1e^{-4}]$. Further implementation details can be found in Appendix B.

In Table 2, we present the Top-1 and Top-5 accuracy achieved by networks in these experiments. The Top-1 accuracy refers to the percentage of image samples whose ground-truth category matches the category most confidently predicted by the network. In contrast, Top-5 accuracy refers to the

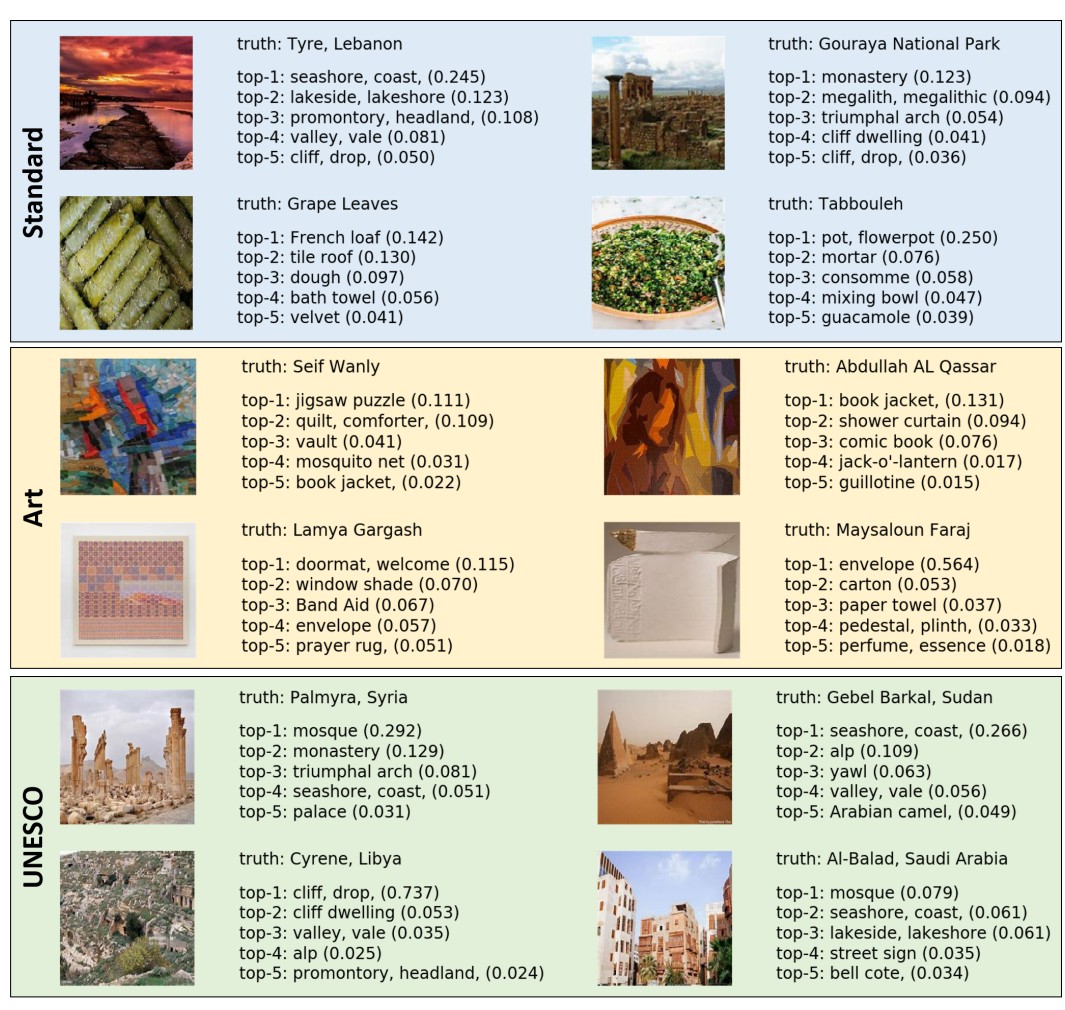

Figure 4: **Top-5 predictions (and confidence) made by an EfficientNet, pre-trained on ImageNet and directly deployed on image samples from the Turath benchmark databases.** We also present the ground-truth *micro* category of each of the image samples. Many of the predictions assign a high probability mass to the incorrect category, lack the finer resolution of our *micro* categories, and do not have a cultural emphasis.

percentage of images samples whose ground-truth category can be found in the Top-5 most confident predictions made by the network[4]. On average, we find that EfficientNet outperforms MobileNetV2 and ResNet50 uniformly across the benchmark databases. For example, on the UNESCO database, EfficientNet, in the linear evaluation phase, achieves Top-1= $39.5$ whereas MobileNetV2 and ResNet50 achieve Top-1= $32.1$ and $33.2$, respectively. We also show that the *micro* category image classification tasks across benchmark databases differ in their level of difficulty. This is evident by the large range of reported accuracy scores. For example, Turath Standard poses the least difficult task with a best Top-1= $46.1$ whereas Turath Art poses the most challenging task with a best Top-1= $16.5$. This is expected given the high similarity of images in the Art database. We believe these accuracy scores, which remain relatively lower than those achieved on ImageNet (Top-1=$90.2$), stand to benefit from further advancements in neural architecture design, transfer learning, and domain adaptation. We also find that fine-tuning networks, regardless of the architecture, is more advantageous than a linear evaluation of such networks. This suggests that the fixed features extracted from a network pre-trained on ImageNet are relatively constraining.

---

[4]We provide demos of these networks in action at `danikiyasseh.github.io/Turath/[benchmark]` `Demo` where benchmark ∈ [Standard, Art, UNESCO].

Table 2: **Image classification test accuracy on the Turath Standard, Art, and UNESCO benchmark databases.** Results are averaged across five random seeds and standard deviation is shown in brackets. Bold results reflect the best-performing network architecture in each benchmark.

| Architecture | Standard (macro) | | Standard (micro) | | Art | | UNESCO | |
|---|---|---|---|---|---|---|---|---|
| | Top-1 | Top-5 | Top-1 | Top-5 | Top-1 | Top-5 | Top-1 | Top-5 |
| *Linear evaluation* | | | | | | | | |
| MobileNetV2 | 70.1 (0.7) | 96.8 (0.1) | 39.1 (0.1) | 62.6 (0.1) | 12.7 (0.2) | 22.4 (0.2) | 32.1 (0.4) | 53.6 (0.2) |
| EfficientNet | **71.2** (0.3) | 96.6 (0.1) | **46.1** (0.2) | **69.5** (0.1) | **16.5** (0.3) | **25.2** (0.3) | **39.5** (0.4) | **60.6** (0.2) |
| ResNet50 | 69.7 (0.2) | 96.9 (0.2) | 39.6 (0.5) | 63.4 (0.3) | 13.2 (0.2) | 23.2 (0.3) | 33.2 (0.3) | 54.0 (0.2) |
| *Fine-tuning* | | | | | | | | |
| MobileNetV2 | 65.6 (1.9) | 95.6 (0.3) | 41.7 (1.2) | 65.9 (1.3) | 12.9 (0.6) | 23.6 (0.6) | 34.4 (0.7) | 56.1 (0.7) |
| EfficientNet | **77.2** (0.6) | **97.6** (0.0) | **49.9** (0.3) | **73.8** (0.3) | **19.0** (0.3) | **31.2** (0.4) | **43.2** (0.4) | **64.2** (0.7) |
| ResNet50 | 71.4 (0.7) | 96.8 (0.1) | 41.2 (1.3) | 65.9 (1.0) | 14.2 (0.8) | 25.0 (1.1) | 35.7 (1.7) | 56.7 (1.4) |

To gain better insight on the type of misclassifications committed on Turath Standard, we present, in Fig. 5 (left), the confusion matrix of macro-category predictions made by EfficientNet on image samples in the test set of the Turath Standard benchmark. This is complemented by Fig. 5 (right) in which we illustrate the UMAP embedding of the penultimate representations ($\mathbb{R}^{640}$) of the same set of image samples. We chose the fine-tuned EfficientNet for these visualizations given its superior performance (see Table 2). In light of Fig. 5, we find that the network is capable of comfortably distinguishing between macro categories. This is evident by the relatively darker diagonal elements in the confusion matrix and the high degree of category-specific separability of the UMAP embeddings. On the other hand, we find that images in the Food category are occasionally misclassified as Dessert, an error which makes sense given the semantic proximity of these categories.

Having shown that an EfficientNet can adequately learn to distinguish between the various categories in the Turath benchmark databases, we wanted to explore whether its classifications were inferred from the appropriate components of the input image. To do so, we exploit an established deep neural network interpretability method, Grad-CAM [27], which attempts to identify the salient regions of the input image in the form of a heatmap. Even though saliency methods have come under scrutiny [28], we find that, in practice, they can be insightful. In Fig. 6, we illustrate the Grad-CAM-derived heatmap overlaid on the original input image presented to a trained EfficientNet alongside the ground-truth annotation of the image. In the case of Leptis Magna (Fig. 6c), we see that the ancient Carthaginian arches are appropriately identified.

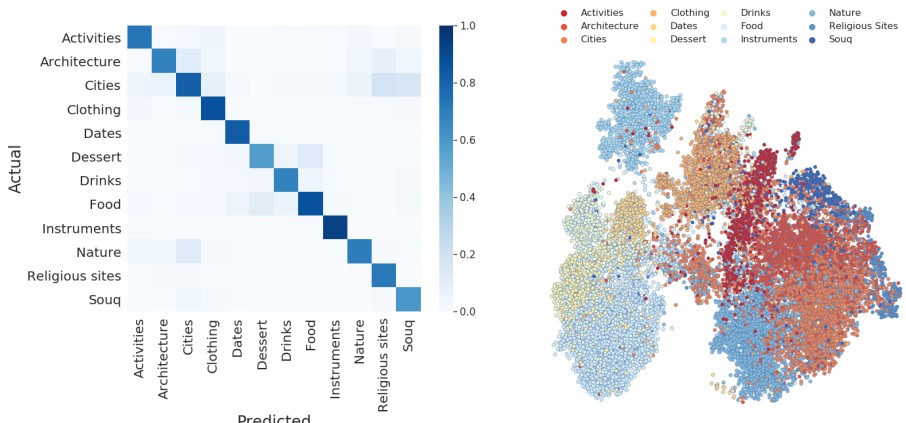

Figure 5: **Performance of EfficientNet fine-tuned on the Turath Standard benchmark database. (Left)** Confusion matrix of predictions made on the test set of the Turath Standard benchmark database. Normalization is performed across columns. **(Right)** UMAP embedding of the penultimate layer representations ($\mathbb{R}^{640}$) of image samples in the test set. We find that the representations exhibit a high degree of separability amongst the macro categories.

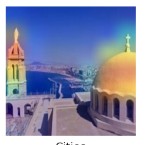 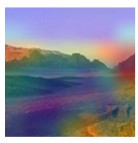 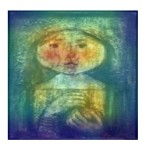 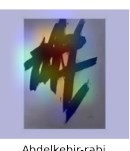 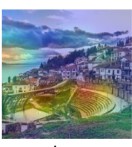 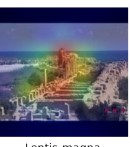

Cities      Nature      Nazir-nabaa    Abdelkebir-rabi    Assur    Leptis-magna

(a) Turath Standard      (b) Turath Art      (c) Turath UNESCO

Figure 6: **Heatmap of the most pertinent regions of the image for the category prediction.** We used Grad-CAM with an EfficientNet trained on the Turath (a) Standard, (b) Art, or (c) UNESCO benchmark databases. Red and blue regions are of high and low importance, respectively. We see that the network is able to identify regions in the image appropriate to the image category.

# 6 Discussion

In this paper, we discussed how existing image databases under-represent objects, activities, and scenarios commonly found in certain cultures. To increase the cultural diversity of image databases, we introduced Turath, a database of approximately 150K images of Arab heritage. Moreover, we proposed three specialized benchmark databases, Turath Standard, Art, and UNESCO, that reflect a range of entities within the Arab world and evaluated several deep networks on such benchmarks. Of the networks evaluated, we found that EfficientNet performed best achieving Top-1 accuracy of 49.9, 19.0, and 43.2, on Turath Standard, Art, and UNESCO, respectively. We hope that our benchmark databases can spur the research community to further advance neural architecture design, transfer learning, and domain adaptation. That being said, it is vital that we consider the limitations and broader societal impact of our work.

**Limitations** When searching for and cleaning the data, we opted out of a crowd-sourcing approach (e.g., Mechanical Turk) in order to scale the database with minimal cost. The machine learning community stands to benefit from the challenge of more independent data cleaning. Despite efforts to clean the data, they exhibit some label noise and may thus benefit from innovative labelling procedures, a challenge we leave to the community. Furthermore, any endeavour dependent on the delineation of categories faces potential biases. Categories simplify and freeze nuanced narratives and obscure political and moral reasoning [8]. Despite our cultural domain knowledge, niche categories that remain undiscovered or unavailable online with sufficient images will not be represented in our database. We aim to continue to engage with artists and heritage specialists to improve the representativeness of our categories.

**Ethics and societal impact** Turath was primarily motivated by the need to increase the cultural diversity of image databases, to improve the applicability of neural networks to under-represented regions, and to actively engage researchers in such regions in the field of machine learning. However, the cultural focus of this database may be prone to abuse by, for example, government and private entities looking to delineate and target cultures for nefarious reasons. To mitigate the abuse of our database for commercial purposes, we are releasing it under a CC BY-NC license, allowing researchers to share and adapt the database in non-commercial settings. More broadly, our belief is that by improving the awareness and understanding of cultures from around the globe, we can better appreciate what they have to offer. Moving forward, we envision the Turath initiative expanding in scope to encompass modalities such as text, audio, and video. Such a path can contribute to research on language preservation, speech recognition, and video analysis.

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
