# OpenReview forum: "Turath-150K: Image Database of Arab Heritage"
_NeurIPS.cc/2021/Track/Datasets_and_Benchmarks/Round1 — Submitted to NeurIPS 2021 Datasets and Benchmarks Track (Round 1)_

### Official Review · Reviewer_qd4s · 2021-07-02
**A New Image Dataset of Arab Heritage to Increase the Cultural Diversity of Existing Databases**

**Rating:** 5
**Confidence:** 3
**Clarity:** The paper is generally well-written a…

**Strengths:**

- The proposed dataset is interesting. To the best of my knowledge, it is the first dataset that focuses on the cultural images of Arab heritage.

- The paper is generally well-written, and the construction process and information of proposed data are explained in detail.

- The introduced dataset and benchmark may inspire the release of additional culture-focused databases and advance future work in neural architecture design, transfer learning, and domain adaptation.

**Weaknesses:**

- The authors claim that existing databases remain largely biased towards objects, activities, and scenarios commonly encountered in a small subset of cultures. Will the introduced dataset be biased to the culture of Arab heritage? The paper tries to address the "hidden tail" problem of existing datasets, while the dataset only presents one specific Arab culture but not many other cultures. Its usefulness remains to be further elucidated. For example, can the fine-tuned models solve the OOD samples aside from Arab culture? Besides, what about the fine-tuned model performance on the original ImageNet test set?

- The paper states that existing datasets violate the assumption that samples are from "the wild" and representative of the underlying data distribution. However, the introduced dataset also represents only one culture of Arab heritage.  Does it also violate this assumption? A better dataset may cover more diverse cultures.

- The practical value and applications of the introduced Turath-150K dataset are not clear.

**Additional Feedback:**

- It would be better to include more cultures into the introduced benchmark to make the claim of dealing with OOD samples more convincing, thus making the benchmark more comprehensive.

- Each category of Turath-150K has over $200$ images. Is this number still not enough to ensure a model can achieve a good performance, especially in handling OOD samples?

- The introduced benchmark only includes three baseline methods under two settings. It would be better to incorporate more methods and settings to make the analysis more comprehensive and reveal its solid contribution.

**Correctness:**

Most of the claims in the submission are correct. Please refer to Weaknesses for some of my concerns.

**Documentation:**

There have been sufficient details on data collection and organization. The dataset is available in the provided URL. However, I am not sure about the intended uses and practical applications of the dataset. The maintenance plan information should be provided.  It would be better to release all the necessary source code and training details to support the reproducibility of the proposed benchmark.

**Ethics:**

From my perspective, there are few or no ethical concerns that warrant further discussion or review.

**Relation To Prior Work:**

The paper discussed how this work differs from previous contributions.

**Summary And Contributions:**

This paper introduces a new dataset, Turath-150K, containing images in the Arab world that reflect objects, activities, and common scenarios to enhance the cultural diversity of existing image datasets, such as ImageNet. For benchmarking purposes, the proposed database is split into three distinct subsets: Turath-Standard (diverse objects, activities, and scenarios commonly encountered in the Arab world), Turath-Art (art from the Arab world), and Turath-UNESCO (heritage sites located in the Arab world). In the benchmark, the authors evaluate three methods to show the limitations of deep neural networks pre-trained on ImageNet that they cannot deal with the out-of-distribution (OOD) samples of the Turath-150K database. With the help of data in Turath-150K, the model performance can be boosted.

---

### Official Review · Reviewer_6gEc · 2021-07-05
**Turath-150K: Image Database of Arab Heritage**

**Rating:** 7
**Confidence:** 4
**Correctness:** Yes, the dataset is constructed in a …
**Clarity:** Yes

**Strengths:**

This paper targets the under-represented domain in the Arab World by providing a large scale dataset centering around this domain. The dataset can be used toward addressing issues such as AI fairness and biases, and potentially can have a large societal and ethical impact in the research community. This paper could inspire similar future works to emerge and to address other under-represented domain as well.

**Weaknesses:**

Although this dataset is a great step forward towards covering the under-represented worlds, the scale of this paper is relatively small when compared to datasets such as ImageNet. I look forward to seeing the authors increase the dataset scale in future iterations. Also, in addition to the data category presented, I am wondering if the authors have plans to extend the scope of this work to face recognition?

**Additional Feedback:**

Please see above

**Documentation:**

Yes.

**Relation To Prior Work:**

Yes the paper stands out from the previous contributions pretty well.

**Summary And Contributions:**

This paper proposes a novel large-scale dataset that focuses on a currently under-represented domain, namely the images from the Arab World. The authors also show that existing models trained on the ImageNet dataset, when directly applied on the Turath dataset, perform sub-optimally and exhibits a certain level of bias.

---

### Official Review · Reviewer_JJXt · 2021-07-06
**Database needs a lot of filtering before it is ready for public use**

**Rating:** 3
**Confidence:** 3
**Clarity:** Yes, the paper is well written and is…

**Strengths:**

The authors aim to tackle the "hidden tail" problem by introducing a new benchmark dataset which highlights Arab culture and introduces new object and activity categories which aren't present in existing large scale image databases. This is a much needed step towards increasing diversity in datasets. This work would help to engage more researchers in the Arab world and would inspire the creation of culture-specific datasets from under-represented regions, thereby increasing the diversity of the machine learning research community as well.

**Weaknesses:**

**Data quality** - The quality of the dataset could be better. Many of the images in Turath-Standard are not representative of the Arab world. For instance, most of the images under the "Activities/archery sport" category are generic images of archery. Even categories specific to the Arab world, such as "Activities/Ayyala folk dance", include many images of folk dances from across the world (this information is present in the corresponding "img_urls.txt" file). More work is needed to improve the quality of retrieved images.

**Personally identifiable information** - Some of the images in the dataset appear to contain personally identifiable information. For instance, "Activities/Desert Horse Riding Dubai/0.jpg" appears to be a screenshot of an Instagram post depicting a man and a horse, including the username. Many of the images in the "Clothing" category, for instance "Clothing/mauritania fashion men/110.jpg", appear to be selfies. A reverse image search reveals that this image was scraped from "https://www.lovehabibi.com/dating/mauritanian-dating/" and reveals the person's first name, age, religion, ethnicity, relationship status, and current city. This raises concerns about data privacy.

**Bias and offensive content** - Many of the images of cities in Syria depict cities in ruins with cars and buildings on fire. Many of them show armed military forces, and injured or deceased people and children. Needless to say, these are quite disturbing and are in stark contrast to the images of skyscrapers from Abu Dhabi. It is quite possible that models trained on this data will be biased due to this distribution shift.

**Additional Feedback:**

**Turath-Art** - Instead of building a face classifier, it might have been simpler to use an off-the-shelf face detector to search for faces in the images and reject those in which a face was found. The paper doesn't mention any performance metrics for the classifier which makes it hard to evaluate its effectiveness.

**Correctness:**

The dataset requires further review and curation to improve the quality of images. In its current state, it is likely to be of limited use to the broader community due to the quality of retrieved images.

**Turath-Standard** - A non-exhaustive list of discrepancies in the dataset is listed below

1. Architecture - The naming scheme varies across sub-folders. A couple of them have no images ("Khirbet Et-Tannur temple" and "Qusayr _Amra")

2. Cities - Many of the categories such as "Daraa-Syria" contain lots of images of maps. About 10% of images of "Deir-ez-Zor-Syria" appear to show cartoon characters including Sonic the hedgehog. Between them and the maps, there are few images of the city.

3. Dates - A quarter of the images of "Rabbi Dates" are images of people and not dates.

4. Drinks - The "saudi champagne" category has very few images of any drinks but has many images of people wearing champagne colored dresses.

5. Food - The "Quzi" category has no images of food, but mainly has images of text which has the word "quiz" in it.

6. Instruments - The naming scheme varies across sub-folders. Some of the them, such as "table_images" have no search or image URLs.

**Minor issues** - The data split ratios as mentioned in Table 1 are 60:10:30 and not 70:10:20 as mentioned in the accompanying text.

Edit (7/19/21) - Many of the images are duplicates at different resolutions. In many instances, the downloaded images appear to be scaled down to low resolutions like 42x42 when higher resolution images are available at the URLs provided with the dataset.

**Documentation:**

The datasheet could be better documented using a framework such as Datasheets for Datasets. The supplementary material contains a list of micro-categories, but is missing several important sections (see https://neurips.cc/Conferences/2021/CallForDatasetsBenchmarks for details) -
1. Author statement and confirmation of license
2. Hosting and maintenance plan
3. Structured metadata

At first glance, the Turath-Art dataset appeared to be full of images of people which was quite unexpected given that the authors described a classifier-based filtering approach which rejected images with faces. After going through the data loading code, it became clear that the authors provided a pickle file with a list of images which need to be skipped. It would be simpler to upload a dataset with these images filtered out.

The metadata containing image URLs is missing for some categories. For the others, the files are treated as binary blobs instead of plain text files and need to be specifically decoded using UTF-8. The individual URLs don't seem to be clearly separated using new lines. In some instances, the base64 encoded jpeg string is provided instead of the URL.

**Ethics:**

Yes, this dataset contains some instances of images with personally identifiable information. It is not clear if people provided their consent on the collection of this data, even if it is publicly available. It is also not clear how the authors will ensure GDPR compliance. This dataset also contains some violent and disturbing imagery which should be filtered out.

**Relation To Prior Work:**

Yes, this work introduces a dataset which aims to increase the representation of images from the Arab world. This dataset contains images from many classes which are under-represented or absent in existing benchmark datasets. It also has more fine-grained labels in the form of micro-categories.

**Summary And Contributions:**

The authors introduce a new dataset, Turath, that is representative of objects, scenes, and activities commonly found in the Arab world. This dataset aims to address the lack of cultural diversity in image databases which they refer to as the "hidden tail". They demonstrate the limitations of models which are pre-trained on existing databases such as ImageNet and introduce new models which are trained and evaluated on Turath.

---

### Decision · Program_Chairs · 2021-07-26

**Decision:**

Reject

**Comment:**

The paper received mixed reviews, leaning toward rejection. There were no responses from the authors. AC finds that the concerns raised by the reviewers are significant enough to warrant rejection.